# Describing the status of reproductive ageing simply and precisely: A reproductive ageing score based on three questions and validated with hormone levels

Kai Triebner[1,2]*, Ane Johannessen[3,4], Cecilie Svanes[3,4], Bénédicte Leynaert[5], Bryndís Benediktsdóttir[6], Pascal Demoly[7], Shyamali C. Dharmage[8], Karl A. Franklin[9], Joachim Heinrich[8,10], Mathias Holm[11], Deborah Jarvis[12], Eva Lindberg[13], Jesús Martínez Moratalla Rovira[14], Nerea Muniozguren Agirre[15], José Luis Sánchez-Ramos[16], Vivi Schlünssen[17,18], Svein Magne Skulstad[4], Steinar Hustad[1,2], Francisco J. Rodriguez[19], Francisco Gómez Real[1,20]

1 Department of Clinical Science, University of Bergen, Bergen, Norway, 2 Core Facility for Metabolomics, University of Bergen, Bergen, Norway, 3 Department of Global Public Health and Primary Care, Centre for International Health, University of Bergen, Bergen, Norway, 4 Department of Occupational Medicine, Haukeland University Hospital, Bergen, Norway, 5 Team of Epidemiology, Inserm UMR1152, Paris, France, 6 Faculty of Medicine, University of Iceland, Reykjavik, Iceland, 7 Department of Pulmonology—Division of Allergy, University Hospital of Montpellier, Montpellier, France, 8 Allergy and Lung Health Unit, Melbourne School of Population Health, University of Melbourne, Melbourne, Australia, 9 Department of Surgical and Perioperative Sciences, Umeå University, Umeå, Sweden, 10 Institute and Outpatient Clinic for Occupational, Social and Environmental Medicine, Ludwig Maximilians University Munich, Munich, Germany, 11 Department of Occupational and Environmental Medicine, Göteborgs Universitet, Göteborg, Sweden, 12 Department of Respiratory Epidemiology, Occupational Medicine and Public Health, Imperial College, London, England, United Kingdom, 13 Department of Medical Sciences, Respiratory, Allergy and Sleep Research, Uppsala University, Uppsala, Sweden, 14 Servicio de Neumología, Servicio de Salud de Castilla–La Mancha, Albacete, Spain, 15 Epidemiologia, Salud Pública, Departamento de Salud, Gobierno Vasco, Galdakao, Spain, 16 Department of Nursing, University of Huelva, Huelva, Spain, 17 Department of Public Health, Aarhus University, Aarhus, Denmark, 18 National Research Center for the Working Environment, Copenhagen, Denmark, 19 Department of Applied Mathematics, University of Malaga, Malaga, Spain, 20 Department of Gynecology and Obstetrics, Haukeland University Hospital, Bergen, Norway

☙ These authors contributed equally to this work.
* kai.triebner@uib.no

**Data Availability Statement:** The data in the tables of the uploaded manuscript contain all information to completely replicate the reproductive ageing

## Abstract

### Objective

Most women live to experience menopause and will spend 4–8 years transitioning from fertile age to full menstrual stop. Biologically, reproductive ageing is a continuous process, but by convention, it is defined categorically as pre-, peri- and postmenopause; categories that are sometimes supported by measurements of sex hormones in blood samples. We aimed to develop and validate a new tool, a reproductive ageing score (RAS), that could give a simple and yet precise description of the status of reproductive ageing, without hormone measurements, to be used by health professionals and researchers.

### Methods

Questionnaire data on age, menstrual regularity and menstrual frequency was provided by the large multicentre population-based RHINE cohort. A continuous reproductive ageing

score. Data used for the validation of the score, such as hormone measurements and other covariates are sensitive patient information and due to Norwegian ethical and legal restrictions only available upon request to qualified researchers. The institutional body responsible for restricting the data is the European Community Respiratory Health Survey consortium. Requests for data access can be directed to Haukeland University Hospital, 5021 Bergen, Norway, Att. Med. Director Alf H. Andreassen; e-mail: postmottak@helse-bergen; phone: +47 55 97 50 00. Org. nr: 983 974 724.

**Funding:** Kai Triebner has received a postdoctoral fellowship from the University of Bergen. The present analyses are part of a project funded by the Norwegian Research Council (Project No. 228174) as well as part of the Ageing Lungs in European Cohorts (ALEC) Study (www.alecstudy.org), which has received funding from the European Union's Horizon 2020 research and innovation program (Grant No. 633212). The European Commission supported the European Community Respiratory Health Survey, as part of the "Quality of Life" program. Bodies funding the local studies are listed in the supporting information. The funders had no role in study design, data collection and analysis, decision to publish, or preparation of the manuscript.

**Competing interests:** The authors have declared that no competing interests exist.

score was developed from these variables, using techniques of fuzzy mathematics, to generate a decimal number ranging from 0.00 (nonmenopausal) to 1.00 (postmenopausal). The RAS was then validated with sex hormone measurements (follicle stimulating hormone and 17β-estradiol) and interview-data provided by the large population-based ECRHS cohort, using receiver-operating characteristics (ROC).

## Results

The RAS, developed from questionnaire data of the RHINE cohort, defined with high precision and accuracy the menopausal status as confirmed by interview and hormone data in the ECRHS cohort. The area under the ROC curve was 0.91 (95% Confidence interval (CI): 0.90–0.93) to distinguish nonmenopausal women from peri- and postmenopausal women, and 0.85 (95% CI: 0.83–0.88) to distinguish postmenopausal women from nonmenopausal and perimenopausal women.

## Conclusions

The RAS provides a useful and valid tool for describing the status of reproductive ageing accurately, on a continuous scale from 0.00 to 1.00, based on simple questions and without requiring blood sampling. The score allows for a more precise differentiation than the conventional categorisation in pre-, peri- and postmenopause. This is useful for epidemiological research and clinical trials.

## 1. Introduction

Menopause marks the cessation of menstruations and the end of the reproductive part of life [1]. This transition, which occurs in women around 50 years of age, takes on average five years and is a major part of reproductive ageing [2, 3]. It also implies profound hormonal changes: a woman's estrogen levels start to decline, and her gonadotropin levels begin to rise. The result is that her body gradually changes into a non-reproductive state [4, 5]. Epidemiological research traditionally uses categories to describe the reproductive status of a woman, such as *perimenopausal* or *postmenopausal*. These categories are based on arbitrary thresholds; they may be rather heterogeneous and lead to an aggravated interpretation of research findings. As an example, the current consensus, the Stages of Reproductive Ageing Workshop (STRAW) defines late perimenopause as a stage lasting one to three years, during which women experience amenorrhea for 60 days or more [6]. This evidently conglomerates a range of women of different reproductive ages, who would then be jointly analysed in an epidemiological study. Categorization may facilitate communication between clinicians and the public, but it does not reflect the underlying biology and it introduces clear limitations, especially for epidemiological research on women undergoing the menopausal transition [7]. The frequent use of categories can however be understood, as there is to date no single biomarker sufficiently describing menopause [6, 8].

We aim to mathematically describe the status of reproductive ageing and create an easy to replicate reproductive ageing score based on the age of a woman and the number and regularity of menstruations she experiences.

## 2. Materials and methods

We used data from two population-based cohorts, the Respiratory Health in Northern Europe (RHINE) study and the European Community Respiratory Health Survey (ECRHS). Ethical approval was obtained for each centre of the RHINE study and the ECRHS from the appropriate institutional or regional ethics committee and each participant provided informed written consent prior to inclusion in the studies.

### 2.1 RHINE

Bergen: Regional Committee for Medical and Health Research Ethics in Western Norway (2010/759); Reykjavik: National Bioethics Committee of Iceland (VSN 11–121); Uppsala, Umeå, Göteborg: Ethics Committee of Uppsala University (2010/068); Aarhus: not required for questionnaire-only studies; Tartu: Research Ethics Committee of the University of Tartu (209/T-17);

### 2.2 ECRHS

Aarhus: Scientific ethical committee for Region Midtlylland; Albacete: Comité Ético de Investigación Clínica del Hospital Universitario de Albacete; Galdakao: Comité Ético de Investigación Clínica del Hospital de Galdakao-Usansolo; Huelva: Comisión de Ética de Investigación Sanitarias del Hospital Juan Ramón Jiménez de Huelva; Bergen: Regional Committee for Medical and Health Research Ethics in Western Norway (2010/759); Bordeaux, Grenoble, Montpellier, Paris: Comite De Protection Des Personnes (2011-A00013-38); Erfurt, Hamburg: Ethikkommission der Bayerischen Landesärztekammer (Reg Nr. 10015); Uppsala, Umeå, Göteborg: Ethics Committee of Uppsala University (2010/068); Reykjavik: National Bioethics Committee of Iceland (VSN 11–121); Tartu: Research Ethics Committee of the University of Tartu (209/T-17);

### 2.3 Development and validation population

The reproductive ageing score (RAS) was developed using data of subjects participating in the questionnaire-based RHINE study and it was validated in the population of the ECRHS where objective sex hormone measurements were available.

The RHINE study is a longitudinal, international, multi-centre study (www.rhine.nu), which includes seven Northern European centres (Bergen in Norway; Reykjavik in Iceland; Umeå, Uppsala and Göteborg in Sweden; Aarhus in Denmark and Tartu in Estonia). For the current paper, we used data from the most recent wave, carried out between 2010 and 2012 with a response rate of 63% [9]. We used data from 3107 women with a mean age of 52 years (range: 38–66 years) to develop the RAS.

The ECRHS also is a longitudinal, international, multicentre study (www.ecrhs.org) [10, 11]. The validation population includes women from 16 centres in nine countries who participated during 2010 to 2012 (Aarhus in Denmark; Albacete, Galdakao and Huelva in Spain; Bergen in Norway; Bordeaux, Grenoble, Montpellier and Paris in France; Erfurt and Hamburg in Germany; Göteborg, Umeå and Uppsala in Sweden; Reykjavik in Iceland and Tartu in Estonia). The available serum samples were analysed for concentrations of follicle stimulating hormone (FSH) and 17β-estradiol. The database contained 1056 women with a mean age of 55 years (range: 40–67 years). For the seven Northern European centres in ECRHS, participants were also a subsample of the RHINE study. Women who participated in both surveys were included into the validation population but not the development population. We further excluded women from both populations who currently used exogenous sex hormones like

contraceptives or hormone replacement therapy (including intermittent progestin therapy), women being pregnant or breastfeeding at the time of the surveys and women reporting irregular menstruation unrelated to menopause (Fig 1) [6, 12, 13].

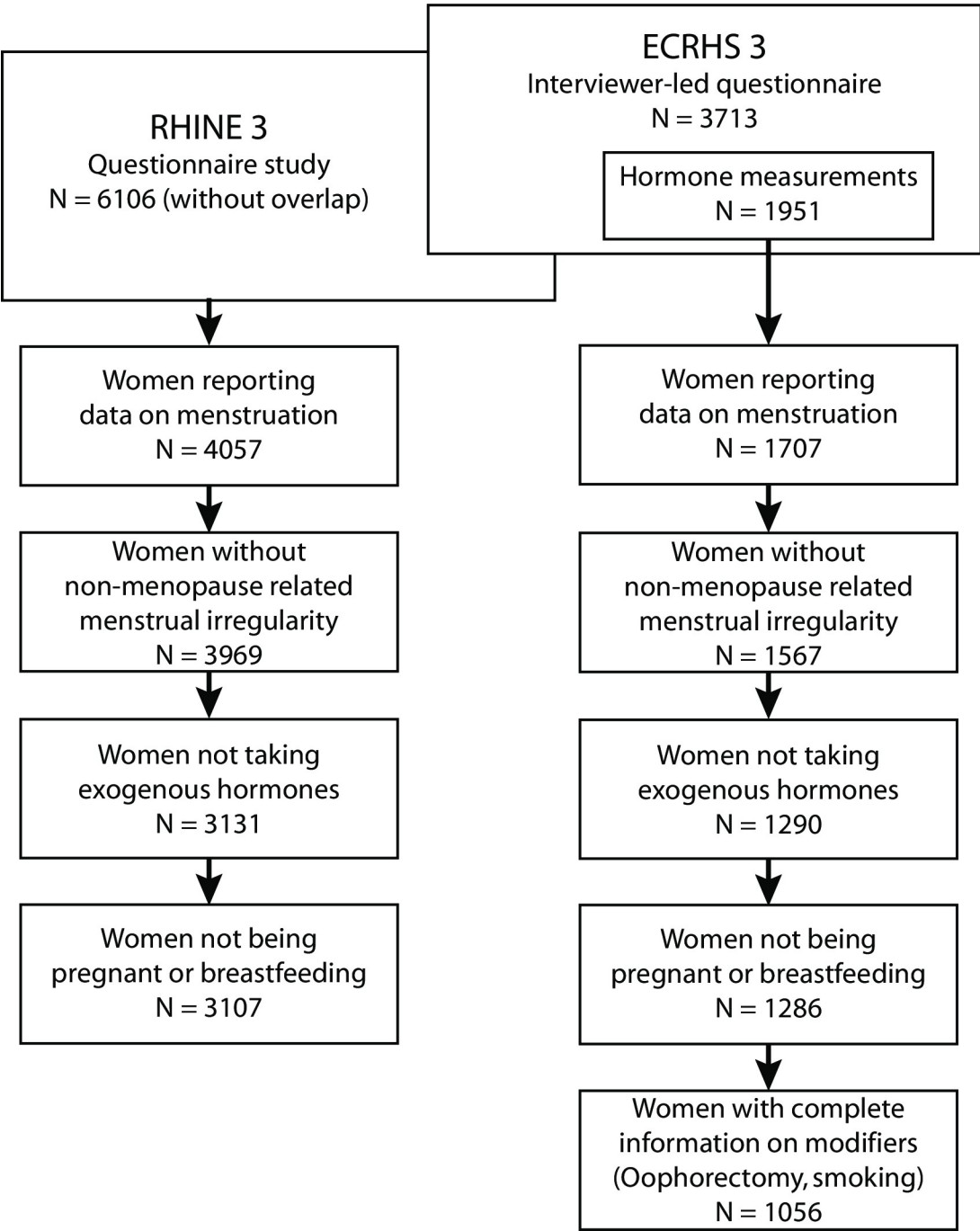

**Fig 1. Flow chart of the development population (left) and validation population (right) with inclusion criteria.** RHINE: Respiratory Health in Northern Europe study, ECRHS: European Community Respiratory Health Survey.

## 2.4 Score development

To develop the score we used techniques of fuzzy set theory, a mathematical concept to depict the biology of physiological processes [14, 15]. It was created in 1964 and successfully implemented in biology, artificial intelligence, and linguistics [14, 16, 17]. Unlike conventional mathematics, which does not allow vague expressions and demands that an object either is a member of a set or not, fuzzy sets are defined by a function (μ) assigning a value between *0.00* and *1.00* to an observation, representing the degree of belonging to a fuzzy set. The value *1.00* means that an object completely belongs to the set and the value *0.00* means the object does not at all belong to the set.

We first defined a function $\mu_A$ based on the number of periods per year and the menstrual regularity. The function requires the proportion *P(period)* of regularly menstruating women for every single number of reported periods as answers to the question: "How may periods did you have in the last twelve months?". A cross-tabulation (Table 1) of the menstrual regularity (answers to the question: "Do you have regular periods?", possible answers: "Yes", "No, they have been irregular for a few months", "No, my periods have stopped") and the number of periods, illustrates the entirety of all values for *P(period)* calculated with Eq 1.

$$P(period) = \frac{x\,(period)}{[x(period) + y(period) + z(period)]}$$

**Equation 1. Proportion of women who have regular menstruation for each number of reported menstruations in the last year** (with period = number of periods per year, x = number of women answering "Yes" to the question: "Do you have regular periods?",

**Table 1. Pattern and number of menstruations in the last year among 3107 women (RHINE).**

| | | "Do you have regular periods?" | | | | |
|---|---|---|---|---|---|---|
| | | Yes | Irregular[a] | No[b] | Total | P (period) [c] |
| Number of periods per year | 0 | 0 | 0 | 1742 | 1742 | 0.000 |
| | 1 | 1 | 1 | 34 | 36 | 0.028 |
| | 2 | 2 | 10 | 26 | 38 | 0.053 |
| | 3 | 3 | 9 | 11 | 23 | 0.130 |
| | 4 | 5 | 18 | 10 | 33 | 0.152 |
| | 5 | 3 | 15 | 7 | 25 | 0.120 |
| | 6 | 6 | 14 | 5 | 25 | 0.240 |
| | 7 | 6 | 24 | 5 | 35 | 0.171 |
| | 8 | 13 | 17 | 6 | 36 | 0.361 |
| | 9 | 13 | 23 | 3 | 39 | 0.333 |
| | 10 | 24 | 28 | 1 | 53 | 0.453 |
| | 11 | 66 | 22 | 1 | 89 | 0.742 |
| | 12 | 779 | 31 | 1 | 811 | 0.961 |
| | 13 | 61 | 6 | 0 | 67 | 0.910 |
| | 14 | 23 | 8 | 0 | 31 | 0.742 |
| | 15 | 12 | 12 | 0 | 24 | 0.500 |
| | Total | 1017 | 238 | 1852 | 3107 | |

[a]"No, they have been irregular for a few months",

[b]"No, my periods have stopped",

[c]Proportion of women with regular menstruation (calculated with Eq 1)

y = number of women answering "No, they have been irregular for a few months" and
z = number of women answering "No, my periods have stopped", e.g. x(11) = number of women reporting regular menstruation among those who report 11 menstruations in the last 12 months).

For women reporting zero menstruations per year this proportion is expected to be zero, as those women are most likely menopausal, while for women reporting twelve menstruations per year it is expected to approach one, as those women are most likely nonmenopausal. For women who report more than twelve menstruations per year it is expected to decline again, as shortening as well as lengthening cycles are an indicator of the beginning menopausal transition [18]. Plotting the complementary proportion *1-P(period)* versus the corresponding number of periods per year, depicts a discrete function, from which the continuous function $\mu_A$ can be approximated using the least squares function approximation.

A second function $\mu_B$ was defined based on age. The construction of $\mu_B$ requires the proportion *P(age)* of women whose menstruations have already stopped for every single reported age. A cross-tabulation of the menstrual status with the reported ages illustrates the entirety of all values for *P(age)* calculated with Eq 2 (Table 2).

$$P(age) = \frac{z\,(age)}{[x(age) + y(age) + z(age)]}$$

**Equation 2. Proportion of women whose menstruations have already stopped, for each reported year of age** (with age = age in years, x = number of women answering "Yes" to the question: "Do you have regular periods?", y = number of women answering "No, they have been irregular for a few months", z = number of women answering "No, my periods have stopped", e.g. x(40) = number of women reporting regular menstruations among those who are 40 years old).

This proportion increases with age and for younger women this proportion is expected to be low. Plotting the proportion P(age) versus the corresponding age depicts a discrete function, from which the continuous function $\mu_B$ can be approximated using the least squares function approximation.

Additionally, two optional modifiers were introduced to $\mu_B$: smoking and unilateral oophorectomy. According to recent literature, current smoking is associated with two years [19] and unilateral oophorectomy with one year [20] younger age at menopause. Therefore, two years were added to the age of current smokers and one year was added to the age of women reporting unilateral oophorectomy, defining the new variable $m_{age}$ (modified age) (Eq 3).

$$m_{age} = \begin{cases} age & for\ non-smokers\ with\ two\ ovaries \\ age+1 & for\ non-smokers\ with\ one\ ovary \\ age+2 & for\ smokers\ with\ two\ ovaries \\ age+3 & for\ smokers\ with\ one\ ovary \end{cases}$$

**Equation 3. Age modification by smoking and oophorectomy**.

Subsequently, the value for the function $\mu_B$ is defined from *P(m_{age})*.

The calculated probability of being not regularly menstruating or without menses, according to the number of periods within the last twelve months ($\mu_A$), does not depend on the woman's age, thus allowing to calculate the probability for each year of being either amenorrheic or not regularly menstruating. Thus, the RAS can be calculated as the union of $\mu_A$ and $\mu_B$ by

**Table 2. Presence of menstruations by age among 3107 women (RHINE).**

| | | "Do you have regular periods?" | | | | |
|---|---|---|---|---|---|---|
| | | Yes | Irregular[1] | No[2] | Total | P(age)[3] |
| Age [y] | 38 | 5 | 0 | 0 | 5 | 0.000 |
| | 39 | 33 | 2 | 2 | 37 | 0.054 |
| | 40 | 86 | 6 | 5 | 97 | 0.052 |
| | 41 | 79 | 10 | 2 | 91 | 0.022 |
| | 42 | 87 | 7 | 5 | 99 | 0.051 |
| | 43 | 99 | 8 | 9 | 116 | 0.078 |
| | 44 | 98 | 14 | 14 | 126 | 0.111 |
| | 45 | 78 | 19 | 19 | 116 | 0.164 |
| | 46 | 86 | 13 | 23 | 122 | 0.189 |
| | 47 | 83 | 17 | 24 | 124 | 0.194 |
| | 48 | 58 | 19 | 28 | 105 | 0.267 |
| | 49 | 47 | 26 | 45 | 118 | 0.381 |
| | 50 | 32 | 27 | 52 | 111 | 0.468 |
| | 51 | 32 | 20 | 77 | 129 | 0.597 |
| | 52 | 27 | 23 | 93 | 143 | 0.650 |
| | 53 | 13 | 14 | 101 | 128 | 0.789 |
| | 54 | 8 | 8 | 122 | 138 | 0.884 |
| | 55 | 9 | 0 | 135 | 144 | 0.938 |
| | 56 | 3 | 3 | 116 | 122 | 0.951 |
| | 57 | 2 | 1 | 131 | 134 | 0.978 |
| | 58 | 6 | 0 | 129 | 135 | 0.956 |
| | 59 | 7 | 0 | 120 | 127 | 0.945 |
| | 60 | 12 | 0 | 115 | 127 | 0.906 |
| | 61 | 8 | 1 | 131 | 140 | 0.936 |
| | 62 | 9 | 0 | 158 | 167 | 0.946 |
| | 63 | 6 | 0 | 96 | 102 | 0.941 |
| | 64 | 4 | 0 | 86 | 90 | 0.956 |
| | 65 | 0 | 0 | 13 | 13 | 1.000 |
| | 66 | 0 | 0 | 1 | 1 | 1.000 |
| | Total | 1017 | 238 | 1852 | 3107 | |

[1] "No, they have been irregular for a few months"

[2] "No, my periods have stopped"

[3] Proportion of women without menstruation (calculated with Eq 2)

adding the two sets ($\mu_A + \mu_B$) and subtracting the overlap, i.e. the repeated elements of the intersection ($\mu_A\mu_B$):

$$RAS = \mu_A + \mu_B - \mu_A\mu_B$$

**Equation 4. The reproductive ageing score as an aggregation function of $\mu_A$ and $\mu_B$.**
The union of $\mu_A$ and $\mu_B$ can be imagined within a three-dimensional Cartesian coordinate system, where $\mu_A$ is projected on the planes parallel to the one spanned by the x- and z-axis for each increment of $\mu_B$ and is projected on the planes parallel to the one spanned by the y- and z-axis for each increment of $\mu_B$. Subsequently the function, representing the union of $\mu_A$ and $\mu_B$ is formed by the maximum value of either function ($\mu_A$ or $\mu_B$) for any given coordinates of

the plane spanned by the x- and y-axis. Thus, Eq 4 can be applied for any woman, knowing the following variables: age, number of periods in the last year, age, oophorectomy ("Never", "One ovary", "Both ovaries") and the smoking status ("Yes" / "No").

## 2.5 Validation of the developed score

We used receiver operating characteristics (ROC) to validate the calculated score against established hormone cut-offs in our validation population (ECRHS) [21] (nonmenopausal: FSH ≤20IU/L and 17β-estradiol ≥147pmol/L, postmenopausal: FSH ≥80IU/L and 17β-estradiol ≤73pmol/L [22]. ROC curves show the true positive rate (sensitivity) against the false positive rate (specificity) and compare to random guessing. An area under the curve (AUC) of one would indicate perfect performance. We validated two concepts. First, we tested how well the score separated nonmenopausal women from the remaining (perimenopausal and postmenopausal) women and second, how well the score separated postmenopausal women from the remaining (nonmenopausal and perimenopausal) women. As perimenopausal women score by definition intermediate values, they were evaluated by comparing intermediate cut-offs of the score to hormone levels and menstrual pattern. For the sake of completeness, this was also done for nonmenopausal and postmenopausal women. Additionally, in order to visualize the performance of the RAS, we include a boxplot of the RAS against three commonly used categories defined by hormonal measurements in the supporting information (S1 Fig).

The least square approximations to develop the RAS were performed using the Maxima CAS (Computer Algebra System) software [23]. All other calculations, including the validation of the RAS, were performed using the R statistical package [24].

## 3. Results

The cross-tabulated data to derive the first function $\mu_A$ based on *P(period)* is presented in Table 1. This function can be approximated by a biquadratic exponential function with a mean-squared error (MSE) of 0.011 (Eq 5). It shows, consistently with existing literature, that the transition to menopause is characterized by an increased frequency of both very long and very short cycles (Fig 2) [18].

$$1 - \mu_A = e^{-0.00047(period-0.7)^4 + 0.009(period-0.7)^3 - 0.0307(period-0.7)^2 + 0.086(period-0.7) - 2.317}$$

**Equation 5. Biquadratic exponential function $\mu_A$ depending of the number of periods.**

The tabulated data to derive the second function $\mu_B$ (Eq 6) based on *P(age)* is presented in Table 2.

The calculated proportion of women who indicate being menopausal by answering "No" to the question "Do you have regular periods?" of all women of the same age ($\mu_B$) can be approximated by a quadratic logistic function (Fig 3) with a MSE for this approximation of 0.002.

$$\mu_B = \frac{e^{(0.0047\ age^2 - 0.0866\ age - 7.646)}}{(1 + e^{(0.0047\ age^2 - 0.0866\ age - 7.646)})}$$

**Equation 6. Quadratic logistic function approximating the function $\mu_B$ (with age in years).**

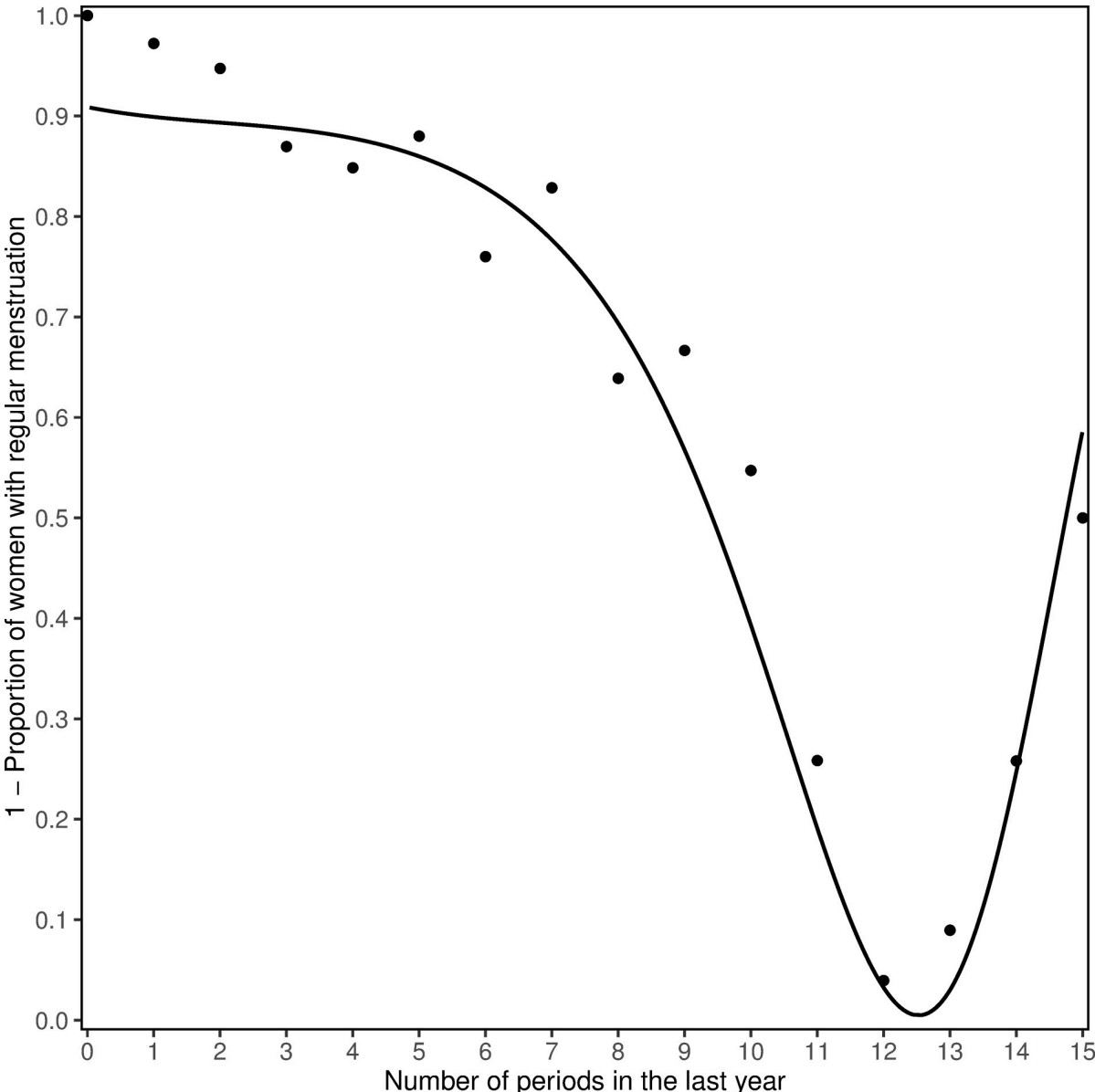

**Fig 2. Approximated function for menstrual regularity.** Data points: Inverse proportion of women with regular menstruation for every response to the number of periods during the last year, observed in the RHINE dataset *1-P(period)*; Line: biquadratic exponential function $\mu_A$ with best fit to observed values *1-P(period)*.

Finally, the function of the RAS (Eq 7) according to the aggregation expressed in Eq 4:

$$RAS = \left(1 - e^{-0.00047(period-0.7)^4 + 0.009(period-0.7)^3 - 0.0307(period-0.7)^2 + 0.086(period-0.7) - 2.317}\right) + \frac{e^{(0.0047\ age^2 - 0.0866\ age\ -7.646)}}{\left(1 + e^{(0.0047\ age^2 - 0.0866\ age\ -7.646)}\right)} - \left(1 - \right.$$

$$\left. e^{-0.00047(period-0.7)^4 + 0.009(period-0.7)^3 - 0.0307(period-0.7)^2 + 0.086(period-0.7) - 2.317}\right) * \frac{e^{(0.0047\ age^2 - 0.0866\ age\ -7.646)}}{\left(1 + e^{(0.0047\ age^2 - 0.0866\ age\ -7.646)}\right)}$$

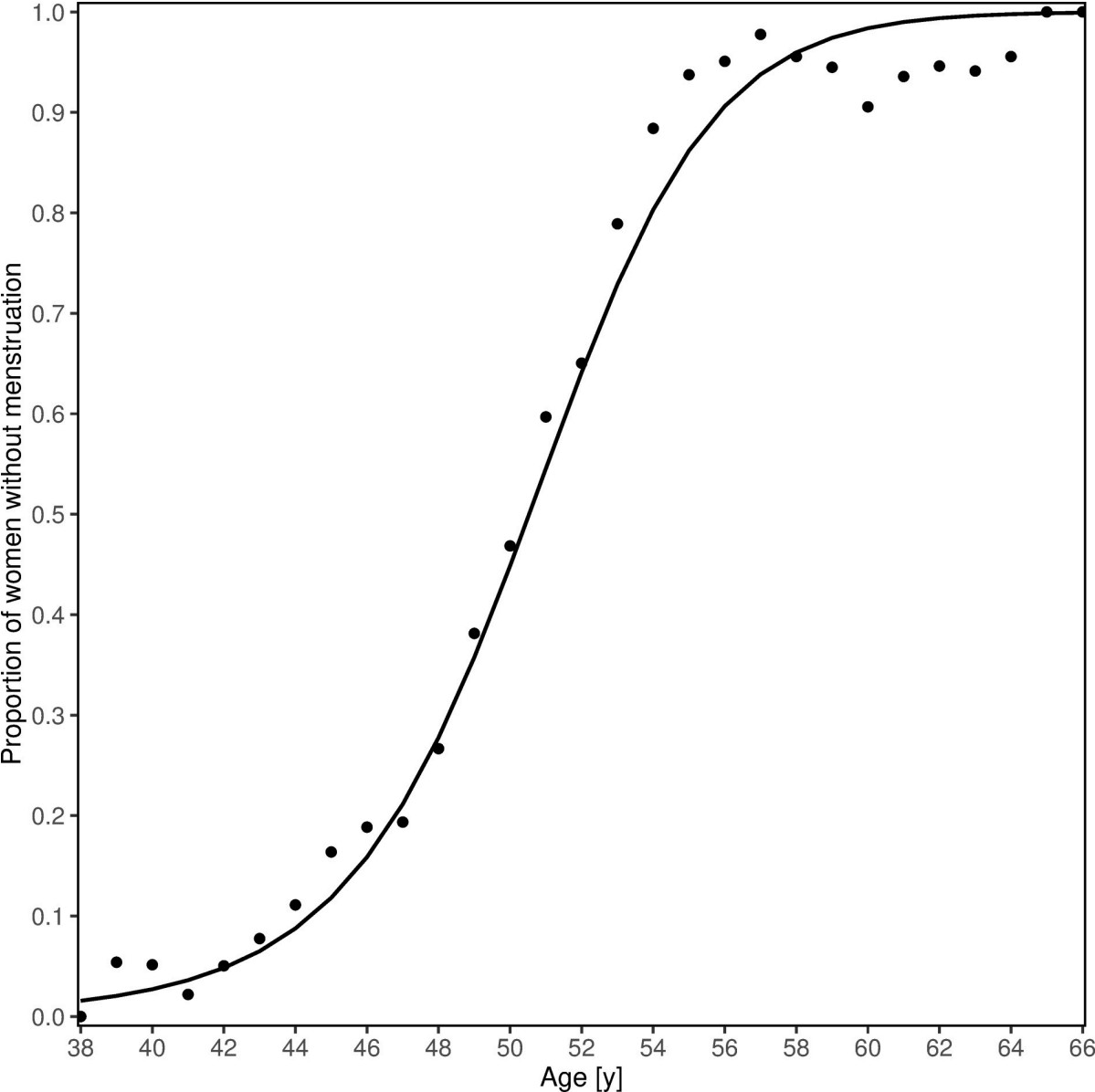

**Fig 3. Approximated function for age.** Data points: Proportion of women without menstruations according to age observed in the RHINE dataset *P(age)*; Line: quadratic logistic function $\mu_B$ with best fit to observed values *P(age)*.

**Equation 7. Final formula to calculate the reproductive ageing score (RAS)** (with period being the number of periods per year and age as the age in years, modified according to smoking status and oophorectomy).

Graphically the RAS can be represented as a three-dimensional figure (Fig 4) using the variables *period* along the x-axis and *age* along the y-axis.

Both, older age and fewer menstruations contribute to a higher RAS and indicate progression into a postmenopausal state. Smoking and oophorectomy act as modifiers of the function $\mu_B$ and thus the RAS. We calculated the RAS for women within the validation population (ECRHS) with Eq 7, which was derived from the development population (RHINE).

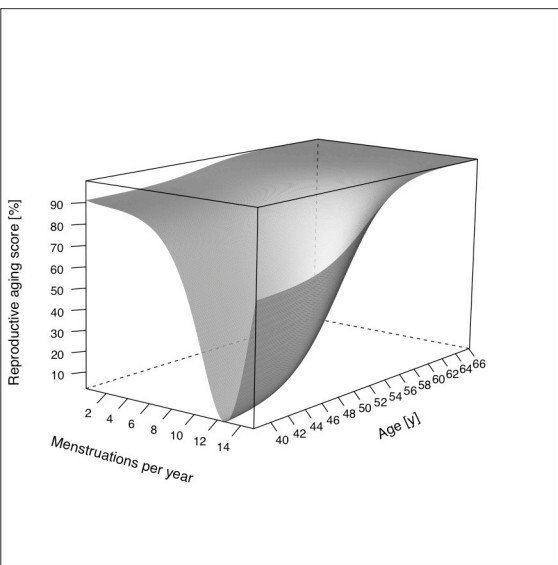

**Fig 4. Unique function of the reproductive ageing score (RAS) (with menstruations per year on the x-axis, age on the y-axis and the RAS, expressed as percentage on the z-axis).**

The area under the ROC curve was 0.91 (95% CI: 0.90–0.93) to distinguish nonmenopausal women from the remaining perimenopausal and postmenopausal women (Fig 5, black curve) and 0.85 (95% CI: 0.83–0.88) to distinguish postmenopausal women from the remaining non-menopausal and perimenopausal women (Fig 5, grey curve), as defined by concentrations of FSH and 17β-estradiol.

Table 3 shows reproductive characteristics for quartiles of the RAS, to illustrate the reproductive characteristics of women with intermediate reproductive ageing scores (0.26–0.75), who are presumed to be in the menopausal transition.

## 4. Discussion

We developed a continuous reproductive ageing score based on age and number of menstruations. Calculating this score for each woman in the validation population (ECRHS) showed that women with the lowest scores featured nonmenopausal characteristics [21, 25]. Women who scored intermediate values showed characteristics of advancing degrees of the menopausal transition and women who scored highest strongly resembled typical postmenopausal women [21, 25]. The RAS can be interpreted as an indicator of the progress of reproductive ageing. Women with a score of 0.00 can be considered premenopausal and women with a score of 1.00 can be considered postmenopausal, while the intermediate values can be considered advancing degrees of reproductive ageing in terms of decreasing fertility, respectively depletion of the ovarian reserve.

The validation with ROC curves and the practical example (supporting information) presented the RAS as a useful tool for epidemiologists. The performance of the RAS for women who were either nonmenopausal or postmenopausal was very good, with AUC values of 0.91 and 0.85, respectively. The ROC curve validation shows that the RAS discriminates nonmenopausal women slightly better than postmenopausal women. The score was able to quantify degrees of the menopausal transition, which has so far not been possible, resulting in women at different stages of reproductive ageing being defined as one heterogeneous group. This tool has great potential to offer new insights into health and disease, e.g. whether women are

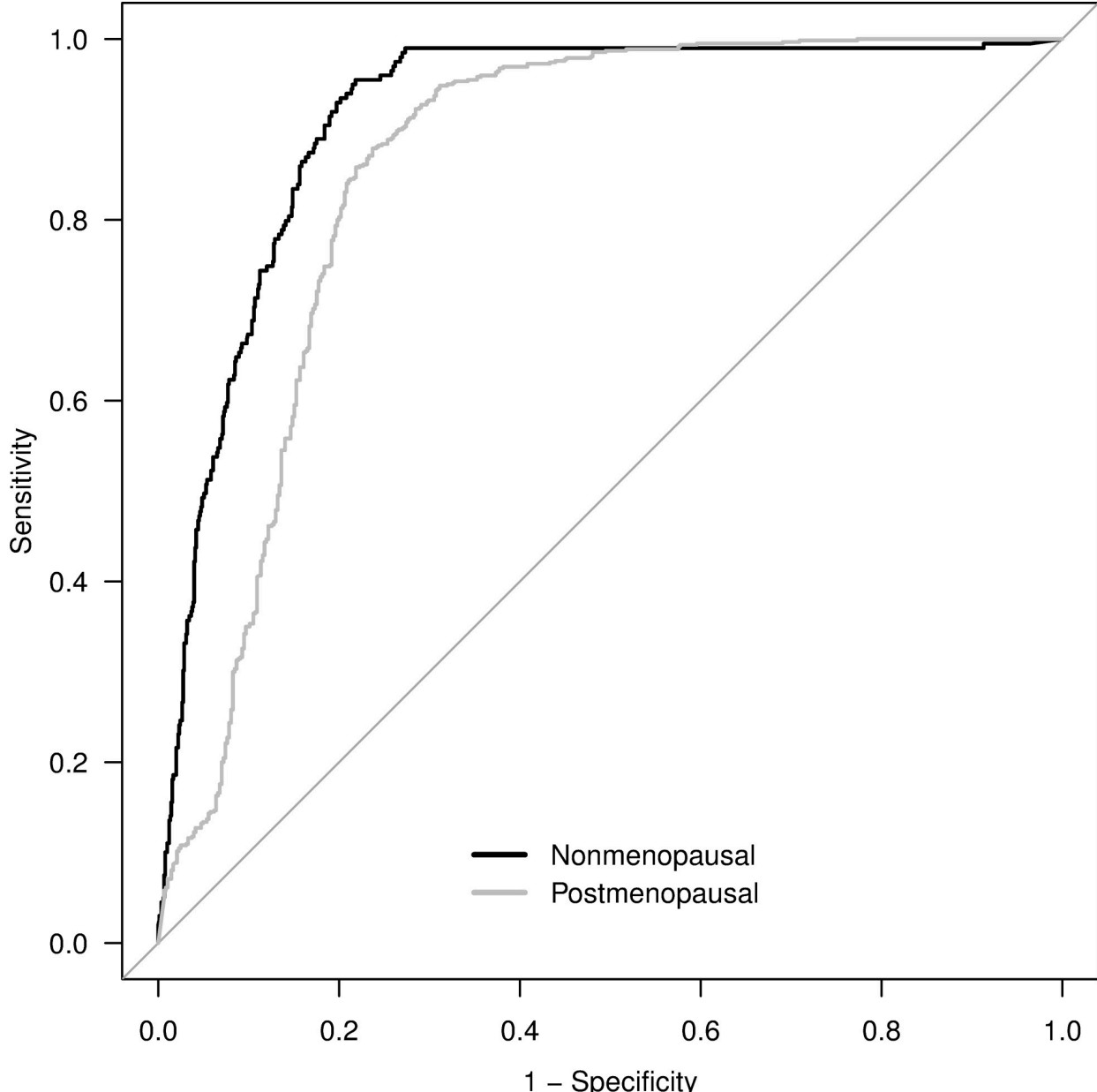

**Fig 5. Receiver operating characteristic for validation of the reproductive ageing score by combined FSH and 17β-estradiol cut-offs in the validation population (ECRHS).** Black curve: Nonmenopausal women versus perimenopausal and postmenopausal women; Grey curve: Postmenopausal women versus perimenopausal and nonmenopausal women.

vulnerable to a certain health condition during a narrow time window within the perimenopausal phase.

In addition, the mean square deviation of the calculated functions reaches values very close to zero, implying a strong nonlinear correlation between the independent variables (number of periods in the last year and age) and the proportions, calculated within the development population. Concerning $\mu_A$, possibly less precise data in the upper range of menstruations per year (>12), due to the lower number of data points, is largely being adjusted for by the relation with age ($\mu_B$) after forming the final function (RAS). The RAS is based on the strong

**Table 3. Quartiles of the reproductive ageing score versus age, menstrual and endocrine status in the validation population (ECRHS).**

| Reproductive ageing score (Quartiles) | 0.00–0.25 | 0.26–0.50 | 0.51–0.75 | 0.76–1.00 |
|---|---|---|---|---|
| Age, mean (SD[1]) [years] | 43.6 (1.9) | 47.5 (1.9) | 50.2 (2.4) | 56.7 (5.5) |
| Periods last 12 months, mean (SD[1]) | 12.1 (0.3) | 12.0 (0.7) | 11.7 (1.4) | 1.2 (3.2) |
| Regular menses [%] | 93 | 97 | 75 | 2 |
| Irregular menses [%] | 7 | 3 | 25 | 11 |
| Amenorrhea [%] | 0 | 0 | 0 | 87 |
| FSH, median (IQR[2]) [IU/L] | 11 (7–16) | 17 (9–27) | 22 (11–48) | 124 (83–166) |
| 17β-estradiol, median (IQR[2]) [pmol/L] | 264 (144–380) | 241 (113–368) | 217 (96–337) | 12 (6–26) |

[1]Standard deviation,

[2]Interquartile range

association between age and the changing number of menstrual periods and its major strength is to quantify reproductive ageing continuously and that it is based on answers to a few simple questions. Important factors influencing reproductive ageing are unilateral oophorectomy and current smoking behaviour, both related to a younger age at menopause, which we accounted for as modifiers. In the case that it is desirable to evaluate smoking behaviour and/or unilateral oophorectomy separately or in a different manner, these modifiers may be removed from the calculation of the RAS.

Another strength is that the RAS can be easily used with all common statistical software and spread sheets by applying the final formula (Eq 7) to a dataset. To replicate the development of the RAS (least square approximations) in other settings we recommend using the open-source software Maxima CAS (https://sourceforge.net/projects/maxima/files/), a specialized computer algebra system that yields high precision numerical results and is capable of including and evaluating complex operations, such as Taylor series and Laplace transforms as well as linear algebra tools like matrix operations, which have been used for the current calculations.

The proportions of women with regular menstruation for the various reported number of periods as well as ages correspond well to the general consensus and what has been described in other studies [6, 8, 22]. Both the development population and the validation population are representative for the women in the relevant age groups in Europe [9], thus the external validity is high for Caucasian populations. For other ethnicities, the functions might have to be slightly modified as age at menopause might differ.

A limitation of the RAS is that potential factors affecting reproductive ageing such as chronic disease, gynaecological disorders and the use of exogenous hormones, are not considered. These limitations are however also acknowledged in the STRAW +10 model [26], which, today, is considered to be the gold standard for assessing reproductive ageing.

It must also be noted that a continuous RAS, indicating how far along a woman is on her way from fertile age to menopause (0.00–1.00) should not be confused with menopause scores assessing women's health after menopause or climacteric symptoms [27, 28].

## 5. Conclusion

The RAS provides a new, innovative and useful tool to describe the current status of reproductive ageing accurately, on a continuous scale from 0.00 to 1.00, based on simple questions and without a need for blood measurements. The score allows for a more precise differentiation between women during this period than the current, conventional categorisation into pre-,

peri- and postmenopause. It thus is useful for epidemiological research and in the design of clinical trials, e.g. studies on hormone replacement therapy.

## Supporting information

**S1 Appendix.**
(DOCX)

## Acknowledgments

We thank all participants, field workers and coordinators of the RHINE study and the ECRHS for their efforts as well as Ersilia Bifulco and Sandra Suske from the Core Facility for Metabolomics at the University of Bergen where the hormone measurements were performed. Further we are very grateful for the support of the Department of Applied Mathematics of the University of Málaga (Spain) and we want to thank Elinor Bartle for revising grammar and language.

## Author Contributions

**Conceptualization:** Kai Triebner, Francisco J. Rodriguez, Francisco Gómez Real.

**Data curation:** Kai Triebner, Ane Johannessen.

**Formal analysis:** Kai Triebner, Francisco J. Rodriguez, Francisco Gómez Real.

**Funding acquisition:** Cecilie Svanes, Bénédicte Leynaert, Bryndís Benediktsdóttir, Pascal Demoly, Shyamali C. Dharmage, Karl A. Franklin, Joachim Heinrich, Mathias Holm, Deborah Jarvis, Eva Lindberg, Jesús Martínez Moratalla Rovira, Nerea Muniozguren Agirre, José Luis Sánchez-Ramos, Vivi Schlünssen, Svein Magne Skulstad, Steinar Hustad, Francisco Gómez Real.

**Investigation:** Kai Triebner, Francisco J. Rodriguez, Francisco Gómez Real.

**Methodology:** Kai Triebner, Francisco J. Rodriguez.

**Project administration:** Ane Johannessen.

**Supervision:** Francisco Gómez Real.

**Validation:** Kai Triebner.

**Writing – original draft:** Kai Triebner, Francisco J. Rodriguez, Francisco Gómez Real.

**Writing – review & editing:** Ane Johannessen, Cecilie Svanes, Bénédicte Leynaert, Bryndís Benediktsdóttir, Pascal Demoly, Shyamali C. Dharmage, Karl A. Franklin, Joachim Heinrich, Mathias Holm, Deborah Jarvis, Eva Lindberg, Jesús Martínez Moratalla Rovira, Nerea Muniozguren Agirre, José Luis Sánchez-Ramos, Vivi Schlünssen, Svein Magne Skulstad, Steinar Hustad.

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
