## [Decision Letter · Decision Letter 0]

9 Mar 2020

PONE-D-20-00740

Defining menopausal status simply and precisely - A Reproductive Ageing Score for epidemiologists, clinicians and the general public, based on three questions and validated with hormone levels

PLOS ONE

Dear Dr. Triebner,

Thank you for submitting your manuscript to PLOS ONE. After careful consideration, we feel that it has merit but does not fully meet PLOS ONE’s publication criteria as it currently stands. Therefore, we invite you to submit a revised version of the manuscript that addresses the all points raised during the review process.

We would appreciate receiving your revised manuscript by Apr 23 2020 11:59PM. To enhance the reproducibility of your results, we recommend that if applicable you deposit your laboratory protocols in protocols.io, where a protocol can be assigned its own identifier (DOI) such that it can be cited independently in the future. For instructions see: http://journals.plos.org/plosone/s/submission-guidelines#loc-laboratory-protocols

We look forward to receiving your revised manuscript.

Kind regards,

Krasimira Tsaneva-Atanasova

Academic Editor

PLOS ONE

Journal Requirements:

Reviewers' comments:

Reviewer's Responses to Questions

**Comments to the Author**

1. Is the manuscript technically sound, and do the data support the conclusions?

Reviewer #1: Yes

Reviewer #2: Yes

2. Has the statistical analysis been performed appropriately and rigorously? 

Reviewer #1: Yes

Reviewer #2: No

3. Have the authors made all data underlying the findings in their manuscript fully available?

Reviewer #1: Yes

Reviewer #2: Yes

4. Is the manuscript presented in an intelligible fashion and written in standard English?

Reviewer #1: Yes

Reviewer #2: Yes

5. Review Comments to the Author

Reviewer #1: The authors aim to generate an objective score for menopausal status.

Overall, Im still struggling with the utility of this score. It essentially uses features of the diagnostic criteria for menopause to give a score indicating the likelihood of menopause.

However, the clinical features presented in table 3, suggest that it probably doesn't remove much heterogeneity to the diagnosis from the use of standard assessments or just asking the relevant information.

I appreciate it doesn't claim to have predictive capability, merely being descriptive of current menopausal status,

but is being validated against diagnostic criteria which have the flaws that the authors are trying to avoid.

However, Im unclear outside of the extremes of the range what knowing the score adds. I appreciate the example in research provided to try to demonstrate the utility in a continuous manner, but given that the validation step is arbitrary (and suspect that any number of similar scores using the information that form the diagnostic criteria for menopause could perform similarly), Im not sure how useful score would be for clinical practice nor how valid it is as an intermediary value for research.

Overall, Im supportive of the concept of trying to make an objective diagnosis of menopausal status, but remain sceptical of the score itself.

In any case, its technically fine, so it will be up to the community to see if the use of this score takes off.

It might be useful to add more detailed information about interpretation of the score beyond high and low,

as in what is the conversion from RAS to the chance of menopause. Can the RAS be converted into a probability of menopause, or is that what it already is supposed to denote ie does 0.86 mean an 86% chance of menopause?

If so, this can be clarified for the reader.

Why was the effect of smoking and oophorectomy not evaluated from the same dataset, but rather an arbitrary value from a different set just added ?

Were other potential modifiers assessed? Previous history of anovulatory disorders eg PCOS?

Line 131- Perhaps shortening of cycles in the lead up to menopausal transition might also contribute to this? Reassuring to see very short cycles in table 1 increase the score.

Line 169- despite the aim to move away from the arbitrary categorisations, the cut offs used to validate the score also seem like consensus categorisations without the possibility of nuance. The FSH threshold of >20 seemed to be one that indicated anovulation and need for contraception rather than nonmenopause per se from the reference. A lower level would identify a more pure non-menopausal group.

Furthermore, FSH levels are known to fluctuate- was there possibility of including repeated values over a threshold?

The ESHRE guidance for POI, uses two measurements of 25 on 2 occasions at least 4 weeks apart to diagnosis POI. An FSH of 15 would be considered high in the absence of raised LH, E2 which could suggest periovulatory value. Similarly not all women will have such a high FSH >80 post-menopause and gonadotropins can fall again after a number of years following menopause also.

Can you plot scatterplots of the RAS by the three categorisations non, peri and post to visualise the performance of the score ?

As it was a longitudinal data set, was it possible to assess 'change in the number of periods per year over more than one year' as a factor?

Table 3- the score seems similar until a score more than 0.75 is reached.

It seems that just asking how many periods in the last year would perform just as well as the RAS.

And for a more subtle / nuanced assessment where there is not a clearcut diagnosis then a combination of inhibin B, AMH, E2, FSH, LH, would be more predictive ?

Can the methods add a paragraph or two explaining the concepts behind fuzzy mathematics for the non-expert reader?

Reviewer #2: The authors address a frequent issue when analysing epidemiological data dealing with age at menopause. The idea of a continuous quantitative score, the RAS, based of simple measurements in any field study is attractive. Age at Menopause has indeed a indirect restrospective definition. The cessation of menses for more than 12 months does not always imply menopause and both FSH and Estradiol level are just mimicking what the gynaecologist can observe directly.

Major Comment:

The developement of the composite score needs to be more carefully explained. Although the calculation of mu_A and mu_B seem easy to understand, allowing to replicate the figures in the last column in table 1 and 2, the merging of the two indices is very briefly noted as obvious like the probability of the union of two sets with a non- zero interception. Their is one basic assumption which may be obvious for the authors but absolutely not for the reader. My understanding is that for each year of age, the authors model the fraction of the women who were not regularly menstruating x times over the last 12 months....The later information is derived from all the women, whatever their ages are.

Minor comments:

- The authors propose a correction for smoking and unilateral oophorectomy based on 2 publications. It has the merit of simplicity but itcould have been compared with models integrating these two covariates into the modeling of the log(mu_A) or the logit(mu_B)

- The referee understands that the shape on figure 2 dictated a transformation that could cope with the strong non linearity with the constraints of staying within the range (0,1). Polynomial regression on the number of periods over the last year was an option, highly dependent on the data available in the range 12+, which are the less precise ones or may have different significances according tho the woman's age, although they profoundly influence the highest degree of the selected polynomes. Did the authors try to compare their model with other models? Some discussions could be added.

The study largely benefit of two large, partly overlaping sets of data from mainly caucasian women from the northern part of Europe. The authors indicate the exclusion of women either using contraceptive hormones, or being pregnant, or breastfeeding. How many women were exposed to intermittent progestin therapy, a common prescription at this period of life in European women? The two flowcharts seem to indicate that they have been also excluded?

Based on these two datasets, the validation study gives very good results at least for caucasian women as correctly pointed out by the authors. The appropriate use of the proposed RSA assumes the availability of similar datasets corresponding to different homogeneous groups of women. If m_B data appear rather easy to get by simple questionnaires, m_A data may be more difficult to obtain.

- The final example on the association between the RSA and the Odds ratio of developint new- onset asthma brings additional value to the proposed score. It is worth noting the large confidence intervals in the late post-menopausal on the figure displaying the results with the traditional approach as compared with the ones on the predicted probability of new- onset asthma in relation to RAS. The comparison is tricky as the abscissa and the ordinates concern different entities and the regression methods is totally different. The shape of the confidence envelope partly reflects the complexe non linearity of the transformation, but the rather "thin" right side remains intringuing. How was the confidence interval calculated?

6. PLOS authors have the option to publish the peer review history of their article (what does this mean?). If published, this will include your full peer review and any attached files.

Reviewer #1: No

Reviewer #2: Yes: Jean-Christophe Thalabard

---

## [Author Response · Author response to Decision Letter 0]

14 Apr 2020

Please see the attached Rebuttal letter.

---

## [Decision Letter · Decision Letter 1]

1 May 2020

PONE-D-20-00740R1

Describing the status of reproductive ageing simply and precisely - A reproductive ageing score based on three questions and validated with hormone levels

PLOS ONE

Dear Dr. Triebner,

Thank you for submitting your manuscript to PLOS ONE. After careful consideration, we feel that it has merit but does not fully meet PLOS ONE’s publication criteria as it currently stands. Therefore, we invite you to submit a revised version of the manuscript that addresses the points raised during the review process.

I would like to the outstanding issues raised by Reviewer 2 addressed in the revised manuscript.

We would appreciate receiving your revised manuscript by Jun 15 2020 11:59PM. To enhance the reproducibility of your results, we recommend that if applicable you deposit your laboratory protocols in protocols.io, where a protocol can be assigned its own identifier (DOI) such that it can be cited independently in the future. For instructions see: http://journals.plos.org/plosone/s/submission-guidelines#loc-laboratory-protocols

We look forward to receiving your revised manuscript.

Kind regards,

Krasimira Tsaneva-Atanasova

Academic Editor

PLOS ONE

Reviewers' comments:

Reviewer's Responses to Questions

**Comments to the Author**

1. If the authors have adequately addressed your comments raised in a previous round of review and you feel that this manuscript is now acceptable for publication, you may indicate that here to bypass the “Comments to the Author” section, enter your conflict of interest statement in the “Confidential to Editor” section, and submit your "Accept" recommendation.

Reviewer #1: All comments have been addressed

Reviewer #2: (No Response)

2. Is the manuscript technically sound, and do the data support the conclusions?

Reviewer #1: Yes

Reviewer #2: Yes

3. Has the statistical analysis been performed appropriately and rigorously? 

Reviewer #1: Yes

Reviewer #2: Yes

4. Have the authors made all data underlying the findings in their manuscript fully available?

Reviewer #1: Yes

Reviewer #2: Yes

5. Is the manuscript presented in an intelligible fashion and written in standard English?

Reviewer #1: Yes

Reviewer #2: Yes

6. Review Comments to the Author

Reviewer #1: Comments addressed, very wide variety of scores for postmenopausal women even. Utility for epidemiological research noted.

Reviewer #2: The revised version adresses most of the comments. However the following suggestions could be taken into account to improve the readibility and the reproducibility of the manuscript

Major Comment:

Although the revised version adds substantial technical information, it does not fully address the basic assumption made by the authors which implies that the probability of either not regularly menstruating or being without menses according to the number of periods within the last 12 months (function mu_A(Period)) does not depend on the woman’s age, allowing to calculate for each year a probability of being either amenorrheic or not regularly menstruating by using the classical formula for the probability of the union of two sets with non null intersection. This is an important assumption which should be made more clear for the reader.

Minor comments:

Comment 12 : The authors propose a correction for smoking and unilateral oophorectomy based on 2 publications. It has the merit of simplicity but itcould have been compared with models integrating these two covariates into the modeling of the log(mu_A) or the logit(mu_B)

Comment on the response 12: We agree with the authors that the proposed score should be easy to calculate justifying their rather simple proposal, but it reincorporates a rather crude categorisation when the authors try to convince the reader to use a continuous variable for quantifying the reproductive aging status.

Comment 13 : The referee understands that the shape on figure 2 dictated a transformation that could cope with the strong non linearity with the constraints of staying within the range (0,1). Polynomial regression on the number of periods over the last year was an option, highly dependent on the data available in the range 12+, which are the less precise ones or may have different significances according tho the woman's age, although they profoundly influence the highest degree of the selected polynomes. Did the authors try to compare their model with other models? Some discussions could be added.

Comment on the response  13: the referee tried to replicate the observed results using the two tables provided in the manuscript. If the logit regression for the age- dependent frequencies gives rather similar results, but not exactly the same ones, for adjusting and graphing the mu_B curve, the adjustment of the mu_A part remains less straightforwards due to the right part of the curve, i.e. number of periods per year above 12. Various methods and selection of the non- linear transform could lead to rather good adjustments. The justification for the least square method as well the reference to the Gaussian Markov theorem has not evident theoretical basis for a regression which concerns either integer values or proportion, although it gives satisfactory results from the practical point of view. My suggestion is to suppress the corresponding sentence without entering into too much irrelevant detail.

Comment 14 : The study largely benefit of two large, partly overlaping sets of data from mainly caucasian women from the northern part of Europe. The authors indicate the exclusion of women either using contraceptive hormones, or being pregnant, or breastfeeding. How many women were exposed to intermittent progestin therapy, a common prescription at this period of life in European women? The two flowcharts seem to indicate that they have been also excluded?

Response 14: OK for the added sentence

Comment 14 bis: based on the two large available datasets, the validation study gives very good results at least for caucasian women as correctly pointed out by the authors. The appropriate use of the proposed RAS assumes the availability of similar datasets corresponding to different homogeneous groups of women. If m_B data appear rather easy to get by simple questionnaires, m_A data may be more difficult to obtain.

Comment on the absence of response to comment 14 bis: It is suggested to add in the supplementary material the R- code for adjusting the datasets, in order to allow epidemiologists to replicate the process in other settings. As an additional comment, it was not clear for the reader how the authors used the Maxima CAS software considering the absence of real theoretical development. A clear statement should be made about the only necessity of the R statistical package to replicate the results.

Comment 15 : The final example on the association between the RAS and the Odds ratio of developint new- onset asthma brings additional value to the proposed score. It is worth noting the large confidence intervals in the late post-menopausal on the figure displaying the results with the traditional approach as compared with the ones on the predicted probability of new- onset asthma in relation to RAS. The comparison is tricky as the abscissa and the ordinates concern different entities and the regression methods is totally different. The shape of the confidence envelope partly reflects the complex non linearity of the transformation, but the rather "thin" right side remains intringuing. How was the confidence interval calculated?

Comment on the response 15 :  

The response remains rather vague for the reader. I understand that the authors performed in the previous published analysis a classical regression analysis of the occurrence of asthma episodes in relation to the menopausal status and perhaps age and here they are performing a second one of the same variable in relation to RAS and perhaps age. I assume that the second one was at least quadratic on RAS to take into account a possible more pronounced effect during the peri- menopausal period. Subsequently, I assume that the authors used some sort of delta method to calculate the confidence interval of the predicted probability from the variance of the regression variable taking into account the var/ cov matrix of the adjusted coefficients of the model. How it ends up with a very thin punctual confidence interval when RAS is 1 is not totally obvious. It could be useful to add more details in the appendix.

7. PLOS authors have the option to publish the peer review history of their article (what does this mean?). If published, this will include your full peer review and any attached files.

Reviewer #1: No

Reviewer #2: Yes: Jean- Christophe Thalabard

---

## [Author Response · Author response to Decision Letter 1]

13 Jun 2020

Please find attached a rebuttal letter with detailed point-to-point responses to the reviewer comments.

---

## [Editor Report · Decision Letter 2]

17 Jun 2020

Describing the status of reproductive ageing simply and precisely - A reproductive ageing score based on three questions and validated with hormone levels

PONE-D-20-00740R2

Dear Dr. Triebner,

We’re pleased to inform you that your manuscript has been judged scientifically suitable for publication and will be formally accepted for publication once it meets all outstanding technical requirements.

Kind regards,

Krasimira Tsaneva-Atanasova

Academic Editor

PLOS ONE
---

## [Editor Report · Acceptance letter]

19 Jun 2020

PONE-D-20-00740R2 

Describing the status of reproductive ageing simply and precisely: A reproductive ageing score based on three questions and validated with hormone levels 

Dear Dr. Triebner:

I'm pleased to inform you that your manuscript has been deemed suitable for publication in PLOS ONE. Congratulations! Your manuscript is now with our production department. 

Kind regards, 

on behalf of

Dr. Krasimira Tsaneva-Atanasova 

Academic Editor

PLOS ONE